# Role of Regular Physical Activity in Neuroprotection against Acute Ischemia

**DOI:** 10.3390/ijms21239086

**Published:** 2020-11-29

**Authors:** Domenico Di Raimondo, Giuliana Rizzo, Gaia Musiari, Antonino Tuttolomondo, Antonio Pinto

**Affiliations:** Division of Internal Medicine and Stroke Care, Department of Promoting Health, Maternal-Infant. Excellence and Internal and Specialized Medicine (PROMISE) “G. D’Alessandro”, University of Palermo, 90127 Palermo, Italy; giulianarizzo@yahoo.it (G.R.); gaiamusiari@gmail.com (G.M.); bruno.tuttolomondo@unipa.it (A.T.); antonio.pinto@unipa.it (A.P.)

**Keywords:** ischemic stroke, physical activity, inflammation, neurotrophins, myokines, neuronal recovery, brain-derived neurotrophic factor (BDNF)

## Abstract

One of the major obstacles that prevents an effective therapeutic intervention against ischemic stroke is the lack of neuroprotective agents able to reduce neuronal damage; this results in frequent evolution towards a long-term disability with limited alternatives available to aid in recovery. Nevertheless, various treatment options have shown clinical efficacy. Neurotrophins such as brain-derived neurotrophic factor (BDNF), widely produced throughout the brain, but also in distant tissues such as the muscle, have demonstrated regenerative properties with the potential to restore damaged neural tissue. Neurotrophins play a significant role in both protection and recovery of function following neurological diseases such as ischemic stroke or traumatic brain injury. Unfortunately, the efficacy of exogenous administration of these neurotrophins is limited by rapid degradation with subsequent poor half-life and a lack of blood–brain-barrier permeability. Regular exercise seems to be a therapeutic approach able to induce the activation of several pathways related to the neurotrophins release. Exercise, furthermore, reduces the infarct volume in the ischemic brain and ameliorates motor function in animal models increasing astrocyte proliferation, inducing angiogenesis and reducing neuronal apoptosis and oxidative stress. One of the most critical issues is to identify the relationship between neurotrophins and myokines, newly discovered skeletal muscle-derived factors released during and after exercise able to exert several biological functions. Various myokines (e.g., Insulin-Like Growth Factor 1, Irisin) have recently shown their ability to protects against neuronal injury in cerebral ischemia models, suggesting that these substances may influence the degree of neuronal damage in part via inhibiting inflammatory signaling pathways. The aim of this narrative review is to examine the main experimental data available to date on the neuroprotective and anti-ischemic role of regular exercise, analyzing also the possible role played by neurotrophins and myokines.

## 1. Methodology of Literature Search

### 1.1. Data Sources and Search

A comprehensive literature search was carried out in the MEDLINE database (search terms: exercise therapy, training, physical fitness, physical activity, rehabilitation, ischemic stroke, neurotrophins; myokines; neuronal damage). The research was conducted for a period between 2000 and 2020. Selected references from some reviews examined were consulted and integrated into this review without limits concerning the year of publication. Medline is certainly one of the most important databases in the world where many researchers rely on it as the first (and often the only) source of literature research. We are confident that the data search carried out on a single database of great quality and international recognition is adequate to ensure an appropriate level of accuracy regarding the number and quality of the articles evaluated. In addition, we sought literature by examining reference lists in original articles and reviews. We have primarily identified systematic reviews and thereafter identified additional controlled trials. We then selected studies in which the intervention was aerobic exercise or strength conditioning and have accorded priority to randomized-controlled trials. Non-controlled trials and controlled trials in which the randomization was uncertain have been included in cases where the other literature was sparse, or where these studies contained important information, for example concerning the form of exercise. The detailed exclusion criteria were (i) studies written in languages other than English, (ii) congress or workshop publications, (iii) animal studies in which the analysis of the methodology used could not reveal applicability of the results to humans, (iv) clinical phase I or II trials (v) studies in which subjects under 18 years of age have been enrolled.

Figure 1 displays a flow diagram explaining the process of the literature search and selection.

### 1.2. Data Analysis

Every author who contributed to the literature search extracted the most pertinent knowledge whilst others verified the accuracy and completeness of the extracted data. Each author made a judgment as to whether the search results were different or confounding in order to release a complete overview of the field.

## 2. Introduction

According to the American College of Sports Medicine (ACSM), Physical Activity (PA) can be defined as any bodily movement produced by the contraction of skeletal muscles that results in a substantial increase in caloric requirements over resting energy expenditure [1]. Contrary to usual belief, exercise is not a synonym rather a subcategory of PA, consisting of planned, structured and repetitive bodily movement done to improve and/or maintain one or more components of PA (e.g., swimming) [2]; all exercise involves some combination of isometric (static) and isotonic (dynamic) stress.

Regular PA has multiple benefits. Several recent meta-analyses show robust epidemiological evidence indicating that regular PA is associated with a reduction of all the main adverse cardiovascular outcomes as well as reduced incidence of type 2 diabetes, hypertension, different types of cancers, osteoarticular disorders [3,4,5]. A risk reduction of coronary heart disease and stroke between 20% and 30% both in men and women in response to a high level of leisure time PA and a possible reduction (10% to 20%) of risk of cardiovascular disease in response to moderate level of occupational PA is reported [6,7].

When prescribed for a medical purpose, exercise is commonly divided into the broad subgroups of aerobic/endurance (i.e., running, walking) and anaerobic/resistance (i.e., weight lifting) activity, although many exercise modalities combine both types. The different types of exercise differ substantially from each other: aerobic and anaerobic exercises involve in a dissimilar way the metabolic patterns, the cardiovascular and respiratory response, the types of muscle fibers used [8] More specifically, as defined by ACSM, aerobic exercise is considered as any activity that uses large muscle groups, can be maintained continuously and is rhythmic in nature; anaerobic exercise has been defined as intense PA of very short duration, fueled by the energy sources within the contracting muscles and independent of the use of inhaled oxygen as an energy source [1]. These remarkable discrepancies between the different types of exercise explain the various chronic adaptations in several organs and systems that can be observed in trained subjects after a quite long period of time.

Various studies have been published that prove the advantages of both aerobic and anaerobic exercise resulting in a positive correlation towards improved cardiovascular health [8]. The peculiar characteristics of aerobic training (AT), or endurance training, lead to beneficial effects on various aspects related particularly to cardiovascular wellness. Studies have shown that AT elicits an adaptive, beneficial form of cardiac remodeling that involves cardiomyocyte growth and proliferation, these effects are directly associated with epigenetic adaptations associated with optimization of oxygen use for ATP production within cardiomyocytes [9]. Furthermore, AT improves oxygen delivery to tissues raising the cardiac output and positively influencing all the mechanisms involved [10]. The increased need for oxygen and energetic substrates leads also to an increase in brain flow [11]; for these reasons AT is widely considered the method of choice for stimulating angiogenesis and inducing the associated local and systemic health-related benefits.

Resistance Training (RT), a form of strength training wherein effort is performed against a specific opposing force generated by resistance, mainly enhances muscle strength, endurance and muscle mass, prevents osteoporosis and sarcopenia in elderly [12]. Moreover, RT exert a beneficial influence on the cardiovascular system, by stimulating the synthesis of molecules such as C-type natriuretic peptide (CNP), an endothelium-derived natriuretic peptide (EDNP) which, through its effects in vascular tone and its antifibrotic and antiproliferative properties may be key elements linking anaerobic RT and improved cardiovascular function [13].

Consensus documents released after independent analysis by leading world organization such as the American College of Sports Medicine and the Centers for Disease Control, the National Institutes of Health (NIH), the US Department of Health and Human Services (HHS) and the European Society of Cardiology (ESC) each concluded that habitual moderate-intensity exercise is an effective tool to reduce the overall risk of chronic disease [14,15,16,17,18]. Nowadays, the ACSM recommends for older adults and patients ≥ 50 years with chronic disorders, a program incorporating aerobic, strengthening, flexibility and balance training. Aerobic training should be done ≥5 days/week for 30 min of moderate intensity or ≥3 days per week for 20 min at vigorous intensity [19]. It should be clearly emphasized that the identification of the optimal dose of exercise, more specifically the exercise dose required to produce the higher level of benefit (exercise dose can be defined as the combination of three determinants: duration, frequency, and intensity), varies considerably across key outcomes of clinical relevance (control of risk factors, primary prevention, secondary prevention, mortality). In clinical practice, this means that the prescription of an exercise dose may best be accomplished by identifying a specific therapeutic target for each individual patient. Without prejudice to the above, in the absence of specific recommendations, the availability may be recommended to improve neurological well-being and counteract brain aging.

## 3. Neuroprotective Effects of Regular Physical Activity

Neurological diseases are multifactorial disorders. Lifestyle together with other factors (genetic profile, inflammation and environment) may have a strong influence on their development and this applies especially to neurodegenerative diseases, such as dementia, becoming more prevalent globally resulting in a constantly increasing cost burden for the global health system [20]. Regular exercise has pleiotropic actions on brain function and plays an important role in the prevention and treatment of many neurological diseases such as vascular dementia, Alzheimer’s Disease, Parkinson’s disease and epilepsy as extensively discussed in many good reviews about the topic [5,21,22,23]. Figure 2 mainly summarizes the main mechanisms through which PA could influence the management of major neurological diseases.

Since the first observations that date back to the last years of the second millennium in which the effects of a six-month aerobic exercise training were compared to stretching training, observing an increased performance in tasks that required a high degree of executive control (e.g., response-compatibility and task switching tasks) only after aerobic PA [24] various studies in humans linked lower aerobic fitness in sedentary older individuals with a reduced cerebral blood flow and decreased cognitive performance [25] and aerobic training with the improved cognitive performance [26]. Higher cortical functions, both in healthy subjects and in subjects suffering from one or more neurological diseases, both acute or chronic, are influenced by the subject’s level of physical fitness, in a direct or indirect (through the benefits induced by exercise on cardiovascular well-being and the improved control of the main cardiovascular risk factors and systemic inflammation) manner [21]. Specific regions such as the anterior cingulate and hippocampus have been identified as those most involved [27]. PA also influences analgesia, sedation, anxiolysis and sense of well-being perceived during and after exercise [28], through the modulation of the endocannabinoid system neurophysiological pathway [29], although other studies suggest that also the mesolimbic system may be related to exercise-induced analgesia and perceived well-being [30]. There is also evidence of the releasing of endogenous opioids in frontolimbic brain regions after exercise, related to the euphoric state after intense PA, the so-called “runner’s high” [31], all these being associated with a positive influence on stress, anxiety and depression [21]. The “antidepressant” effect of exercise can also be attributed to the upregulation of the peroxisome proliferator-activated receptor gamma coactivator 1 alpha (PGC-1α) nuclear mediator in exercised muscles. The overexpression of PGC1α in muscle after regular PA seems able to change the kinurenine metabolism (increasing the conversion of kynurenine to kynurenic acid) thus reducing kinurenine entry into the brain; kinurenine mainly is derived from the catabolism of the essential amino acid tryptophan and high levels of kinurenine are found in people with depression [32].

In view of these valuable epidemiological data supporting the beneficial effects of PA on neurological diseases, some mechanisms have been hypothesized to explain what has been observed. Trying to classify the main, three different mechanisms can be identified: Figure 3 summarizes the main beneficial effects of regular exercise in protecting against major acute and chronic neurological disorders.

### 3.1. Enhancement of Antioxidant Activity

Oxidative stress is an imbalance between reactive oxygen species (ROS) production and elimination [33]. NO is created by the action of nitric oxide synthase (NOS), catalyzed the enzymatic reaction of L-arginine and oxygen, and there are three types of this enzyme, such as endothelial NOS (eNOS), neuronal NOS (nNOS), and inducible NOS (iNOS). In particular, eNOS and nNOS are calcium-dependent, while iNOS is calcium-independent. Furthermore, in many cases, low levels of NO generated by eNOS is physiological, whereas NO concentrations produced by nNOS and iNOS are potentially damaging. The most important ROS implicated are nitric oxide (NO-), superoxide (O^2−^), hydrogen peroxide (H_2_O_2_) and peroxynitrite (ONOO-). ROS are generated from specific enzymatic or chemical reactions; NO-, one of the most important, it is generated by eNOS in endothelial cells, during the physiological functioning of the vessel. The activity of vasodilator hormones increases intracellular Ca_2_, and lead to an elevated eNOS activity and a consequent production of NO-. ROS are able to modulate many classes of genes, such as adhesion molecules, chemotactic factors, antioxidant enzymes and vasoactive substances as a reactive response to many stimuli, like for example the activation of superoxide dismutase and catalase by H_2_O_2_ [34]

The brain only represents almost 20% of the body’s oxygen consumption and foremost is the organ most able to generate free radicals. Additionally, the brain is the tissue most vulnerable to free radical damage, because it includes numerous accumulations of lipids with unsaturated fatty acids and several levels of iron [35]. Oxidative damage has been associated with the poor physiological function of the brain; thus, oxidative stress could be a major element in the determination of age-related brain alterations as well as in the pathophysiology of neurodegenerative diseases.

The cerebral ischemia-reperfusion injury is due to many pathophysiological pathways such as the release of excitotoxic neurotransmitters, free radical damage, intracellular Ca^2+^ storage, lipolysis, neuron apoptosis and neuroinflammation [36,37,38,39,40,41]. Among these pathways, free radical damage is considered one of the main characters during ischemia-reperfusion injury, especially if a revascularization procedure is attempted. The role played by oxidative stress in the pathogenesis of ischemic and reperfusion-related brain damage is demonstrated by several studies [42], being the injury mediated through free radicals and lipid peroxidation [43,44]. However, data on biomarkers of oxidative stress in the hyperacute phase of acute ischemic stroke are to-date very limited. Oxidative stress is able to induce neuronal apoptosis by DNA deterioration, lipid peroxidation, and mutation in the shape and in the function of main proteins, this cascade induces more severe oxidative stress injuries [45], in fact, ROS are also able to modulate some major signaling pathways able to trigger death cell and necrosis such as the protein 53 pathway (p53) or the protein kinase (PK) pathway [46].

Several studies in the last years have demonstrated an inverse correlation between PA degree and oxidative stress status. Physical inactivity significantly increases vascular lipid peroxidation and superoxide production [47], Muscle cells play a main role in exercise-induced ROS generation, thus exerting beneficial or detrimental effects per se or in relation to different tissues. Exercise may be considered a stress factor, which can disrupt the homeostatic balance within the body. After PA, an upregulation of biological systems occurs, which exceeds the previous level of function. A growing body of evidence suggests that this also occurs for the antioxidant enzyme activity. Regular moderate aerobic training, inducing the production of antioxidant agents and DNA repair, could ensure that individuals who exercise have a biological advantage over people with sedentary lifestyle profile. Antioxidant enzymes, such as Superoxide Dismutase, Catalase and Glutathione Peroxidase, are more expressed in trained individuals, although these changes are related to specific training performed and to the physical status of the subjects [48]: although regular moderate aerobic training could be beneficial for oxidative stress, acute and strenuous bouts of aerobic and anaerobic exercise could induce oxidative stress. This two-sided outcome can be explained by the hormesis theory [49] according to which a low dose of an agent that is harmful at high doses, induces an adaptive beneficial effect on the cell or organism [50].

Figure 4 briefly outlines the hormesis theory and its implications in the strengthening of the antioxidant power of the organism.

### 3.2. Anti-Inflammatory Activity

A somewhat similar difference between acute and chronic effects of exercise can be observed when talking about the anti-inflammatory effect. An acute bout of exercise could increase the plasmatic level of the same pro-inflammatory cytokines that normally increase during a stress response and could, whereas regular exercise seems able to upregulate the anti-inflammatory capacity of the organism leading to a reduced level of systemic inflammation estimable by the reduction of the main circulatory inflammatory markers. A relevant amount of evidence appears to confirm that regular exercise is linked to amelioration in biomarkers of systemic low-grade inflammation expressed, for example, by the circulating levels of C Reactive Protein (CRP) [51,52,53].

Vascular inflammation and endothelial dysfunction are sustained by the activation of the endothelial cell by cytokines, oxidized Low-Density Lipoproteins (LDL), and ROS, followed by the increased endothelial expression of Cell Adhesion Molecules (CAMs), such as ICAM-1, VCAM-1, E-selectin, and P-selectin, that are crucial to the recruitment of inflammatory cells to the vessel wall [54]. These molecules can be measured in circulation in their soluble form [55]. PA seems to have a positive impact on the circulating CAMs. Koh et al. noticed a decrease in CAMs’ expression following low-to-moderate intensity aerobic exercise, while high-intensity aerobic exercise seemed to momentarily elevate their expression immediately after the exercise [56], thus leading to leucocyte accumulation. This could be related to changes in the epigenetic regulation of CAMs transcription induced by increased shear stress [57]. Besides this direct influence in CAMs expression, vascular anti-inflammatory effects of PA may be explained also throughout the reduction of agonists of CAM synthesis, i.e., inflammatory cytokines [58], ROS [59], and oxidated LDL [60]. The improvement of the antioxidant capacity of the organism secondary to regular PA, as described above, may also help to reduce the vascular inflammation by interfering with one of the key pathogenetic mechanisms of atherogenesis: the LDL oxidation and the foam cells formation [61].

The systemic anti-inflammatory effect of exercise, confirmed by several lines of evidence, is the result of the combination of different mechanisms: direct and indirect modulation of the activity of immune cells, neuroendocrine changes induced by exercise, reduced visceral fat mass, to mention the main. Acute and regular PA causes plasma release of hormones with certain anti-inflammatory activity such as cortisol and catecholamines and is linked to TH2 switch of leucocytes. An intense debate has driven the discussion about the possibility that the anti-inflammatory effect of exercise can also be mediated (to some extent) by certain activities exerted by myokines or other molecules produced by the muscle during prolonged contraction and released into the bloodstream [53]. The production and subsequent release into circulation of muscular myokines appear to be directly related to the duration and intensity of training, while the anti-inflammatory effect of exercise doesn’t seem to have such a direct association. This thesis is supported by evidence that the production of IL-6 from muscle after an acute bout of PA do not reach consistent concentrations with short durations or low-to-moderate intensity of training [62] and that routine PA determines a decrease of IL-6 levels (instead of IL-10 concentration that is shown to increase). On the other hand, many studies reported data that assess how also low-intensity programs of exercise, such as fast walking [63,64,65,66], could produce a consistent decrease in plasma markers of systemic inflammation although does not determine any enhance in circulating cytokines expression [67]. Finally, it should be pointed out that, in clinical practice, interventions aimed at lifestyle correction often do not only include the implementation of the PA level but also other lifestyle advice such as weight loss, low-fat diet, smoking cessation or also in combination with pharmacological treatments; it cannot be excluded that all these interventions together can have a mutually reinforcing effect in strengthening the anti-inflammatory effect of the exercise.

### 3.3. Improved Neuronal Plasticity

The term “neuroplasticity” is a novel concept referring to the potential ability of the nervous system to modify its structural and functional characteristics answering to altered demands and environments [68]. Neuroplasticity is a dynamic process involving changes in the number of brain nuclei and structures, numerous functions (learning, memory, movement) and various interactions [69]. Evidence from both human and animal studies have suggested that PA has a facilitating effect on neuroplasticity; regular PA stimulates angiogenesis, synaptogenesis, neurogenesis as well as the synthesis of neurotransmitters in various cerebral areas [70,71,72,73,74,75].

PA directly influences neurotransmission by triggering both central [76] and peripheral [77] catecholamine levels (i.e., dopaminergic, noradrenergic, and serotonergic systems): catecholamines produced peripherally lead to increased serum calcium levels via the activation of the calcium-calmodulin system, enhanced by exercise, and, as a consequence, a major transport of the ion into the brain [21].

PA up-regulates genes related to synaptic plasticity such as the N-methyl-D-aspartate receptor (NMDA-R) [78,79]. The majority of the up-regulated genes has a key role in: (i) synaptic function (i.e., synapsin I and II, synaptotagmin, and syntaxin); (ii) signal transduction pathways associated with memory processes (i.e., calcium/calmodulin protein kinase II (CaM-KII), mitogen-activated protein kinase (MAP-K/ERK, I and II), protein kinase C (PK-C)); (iii) transcription factors as cAMP response element-binding protein (CREB) [80,81] which has the main role for long-term neuronal plasticity and for the formation of long-term memory. Exercise affects proliferation, size and function of astrocytes, thus regulating the number and localization of neuronal synapses and influencing the formation of episodic memory and long-term potentiation (LTP) [82], the neural basis of learning and memory [83]. The induction and maintenance of LTP require the involvement of NMDA-R as well; LTP is in fact the result of a synergistic action at the synaptic level between presynaptic neurotransmitter release and postsynaptic membrane NMDA-R channel opening [84,85] On the other hand, glutamate induces neurotoxicity mainly through the over-activation of NMDA-Rs in the post-synaptic membranes of neurons and recent studies have highlighted the involvement of glutamate-induced excitotoxicity in the pathogenesis of stroke [86]. The exercise-induced neurotrophin insulin-like growth factor (IGF)-1 protects cultured hippocampal cells against NMDA-mediated excitotoxicity by modifying the phosphorylation of NMDA_2B_ [87]. Over the past few years, pre-ischemic treadmill training (hence PA) has been proven to inhibit the overexpression of GluN2B (NMDA subunit) and the mRNA expression of metabotropic glutamate receptor 5 (mGluR5) thus reducing brain damage [88].

Erikson et al. [89] studied N-acetylaspartate (NAA) as a putative element influenced by PA and involved in neuronal capability and metabolism. NAA physiologically controls several aspects of the development and interaction of brain cells by acting on cellular metabolism and myelination [90] thus playing a key role in the establishment and maintenance of a normal, functional vertebrate nervous system [91]. Age-related reductions in the concentrations of NAA have been found especially throughout late life, meanwhile, higher NAA levels were associated with better working memory performance; additionally, in older adults, higher aerobic fitness levels offset an age-related decline in NAA giving increased neuronal viability [89]. Finally, an important role in promoting exercise-related synaptic plasticity is played by neurotrophins, including insulin-like growth factor (IGF-1) and brain-derived neurotrophic factor (BDNF) [69]. Rodent models have shown that short-term periods of increased PA are sufficient to upregulate central and peripheral factors that support the brain, and promote synaptic plasticity through increased expression of mRNA for BDNF and IGF-1 but also for Nerve growth factor (NGF), and tropomyosin receptor kinase B (TrkB) [78,92].

IGF-1 might participate in behaviorally induced plasticity [93,94] whereas BDNF, whose effects and mechanism of action will be discussed in more detail below, is thought to be one potential element linking PA with memory, neurogenesis and synaptogenesis.

## 4. Immuno-Inflammatory Activation during and after the Acute Phase of an Ischemic Stroke

Stroke is globally the second cause of death after ischaemic heart disease [20]. Many evidence available indicates that almost 90% of cardiovascular disease, including stroke, is caused by potentially modifiable risk factors [95]. In this perspective, improved risk factor control and a better management scheme along with a supporting role of stroke awareness programs could be a key to decrease stroke rate as well as post-stroke residual disability. Evidence shows that exercise can positively influence several physical (e.g., cardiovascular fitness, walking ability, muscle strength) and psychosocial (e.g., depressive symptoms, memory, fatigue) domains after a stroke [96].

Ischemic stroke has been described as a “thrombo-inflammatory disease”, since thrombosis and inflammation are highly intertwined processes that interact at multiple points as contributors to brain injury and stroke progression [97] Many chronic systemic inflammatory conditions, such as atherosclerosis, diabetes and obesity are associated with increased risk of stroke, which suggests that systemic inflammation may contribute to the development of stroke in humans [98]. These elements tell us that inflammation is part of ischemic stroke pathobiology as well as part of its pathophysiological background, and thus has an all-around pivotal role. Atrial Fibrillation itself, another pivotal risk factor for ischemic stroke [99], carries a separate systemic inflammatory burden, being associated with an increased “inflammatory cytokines milieu” and endothelial dysfunction [100].

Cerebral ischemia initiates a complex cascade of events at genomic, molecular, and cellular levels, and inflammation is important in this cascade, both in the Central Nervous System and in the periphery [101]. The inflammatory response starts locally in occluded and hypoperfused vessels and in the ischemic brain parenchyma, leading to systemic propagation of in situ–produced inflammatory cytokines and to damage in the blood–brain barrier; this creates a route for entry of immune cells, solutes and water into the brain parenchyma resulting in interstitial inflammation and oedema that damages neuronal tissue [102]. The cerebral ischemic injury induces a series of inflammatory events including the infiltration of circulating immune cells (neutrophils and monocytes) and activation of resident cells (i.e., microglia, astrocytes and endothelial cells) [103]. In this complex scenario, cytokines have a central role modulating local changes and systemic host response to Central Nervous System inflammation, infection and injury [101]. Three major cytokines, tumor necrosis factor (TNF-alpha), interleukin (IL)-1, and IL-6 are produced by cultured brain cells after various stimuli such as ischemia [104]. The source of these and other less detectable cytokines remains unclear although serially measured IL-lβ and IL-6 in patients with ischemic stroke, resulted higher in cerebrospinal fluid than in serum, thus being suggestive of intrathecal production of the cytokines [105].

Functions and actions of ischemia-related cytokines in the brain remain to be elucidated, yet probably include both beneficial and detrimental effects [101], suggesting a “multiphasic role” [106]. Inflammation seems in fact to be more damaging during the early phase of ischemic stroke, while in recovery stages it seems to help clear up debris and support wound healing and this could be the reason why anti-inflammatory drugs (steroidal and non-steroidal agents) failed to demonstrate a benefit in thromboembolic infarction [102].

The level of systemic inflammation in stroke patients strongly depends on the etiopathogenesis of the ischemic episode and the whole clinical profile of the patient (number of comorbidities). This different background on which the acute ischemic brain event occurs, makes it more correct to talk about ischemic strokes rather than ischemic stroke [107], since the different diagnostic subtypes are associated with a distinct level of neurological damage and functional impairment as well as with a diverse degree of local and systemic immunoinflammatory activation [98,101,104,108,109,110,111,112,113]. This is clear if one compares, for example, a small lacunar stroke, often subcentimetric, with a cardioembolic stroke, commonly larger and with multiple synchronous lesions, sustained by a continuously active emboligenous heart disease itself associated with high levels of systemic inflammation [100,114].

It is of course conceivable that other factors could represent an additional element influencing immuno-inflammatory activation after acute ischemic stroke, infarct size could represent one additional possible factor influencing immuno-inflammatory marker plasma levels although results in this regard are unclear and more evidence is needed [114,115,116,117].

Finally, ischemic stroke-related inflammatory response is clearly variable. This variability depends, as thoroughly discussed, on stroke subtype (and, consequently stroke’s metabolic and systemic background) and infarct size but could also depend upon genetic polymorphism of candidate genes of inflammatory cytokines, as proinflammatory genetic genotypes (IL-6 GG, ICAM-1 EE, E-Selectin AA) are significantly more common in subjects with stroke history [98,118]. The implicated genes are both pro-inflammatory and/or anti-inflammatory. IL-10 is, for instance, one of these inflammation-related genes implicated in the pathogenesis of ischemic stroke and is a potent anti-inflammatory cytokine whose action drastically limit the size of brain injury, atherosclerosis and inflammatory response, counteracting the effects of pro-inflammatory molecules such as TNF-α; IL-10 (-1082A/G) has been associated with the risk of Large Vessel Disease, Small Vessel Disease, and other subtypes of ischemic stroke, although further studies should be conducted [119].

Understanding the inflammatory background of subtype-related ischemic stroke could have a strong prognostic value. Interestingly, immune-inflammatory activation of the acute phase of ischemic stroke is associated with the severity of the acute neurological deficit evaluated by the National Institutes of Health Stroke Scale (NIHSS) [120]; higher levels of inflammation seem to correlate with a poorer outcome for the patient.

In this regard, our group [114] found that patients with cardio-embolic strokes, in comparison with other subtypes, showed a higher degree of immuno-inflammatory activation as well as a higher degree of acute neurological deficit whereas patients with lacunar stroke, compared to other subtypes, showed a lower degree of immuno-inflammatory activation with a lower degree of acute neurological deficit.

Figure 5 shows the main features of the immuno-inflammatory activation during an ischemic stroke.

## 5. Anti-Ischemic Effects of Physical Activity

With limited therapeutic opportunities, stroke remains a relevant critical disease with multiple hopefully modifiable pathophysiological mechanisms whose better knowledge could lead to a most successful post-stroke rehabilitation program. An ischemic stroke is often triggered by cardio-embolism or athero-thrombotic arterial obstruction causing a sudden decrease in cerebral blood flow. In view of the peculiar terminal circulation that can be found in the brain, a gradient of decreased blood flow emerges, causing severe neuronal injury and blood flow reduction and resulting in a surrounding penumbra (ischemic penumbra) in which degenerative reactions and blood flow reduction are partly reversible. Neuronal damage due to ischemic stroke is attributable mainly to two mutual pathological processes: oxygen loss and disturbance in glucose supply to affected brain areas, triggering the immune-inflammatory cascade previously briefly discussed. The health of the cerebral vessels is therefore a central element of protection against ischemic stroke [122].

It is well known the role carried out by exercise in mediating the recovery of post-stroke damage. PA can contribute to the functional compensation of surviving brain areas involved in limb function: the possible mechanisms include enhanced activity and axonal growth of the pyramidal and extrapyramidal systems in the ipsi-lesional and contralesional hemispheres [69]. Despite the intriguing pathophysiological context and the numerous experimental data available, only a few studies have attempted to ascertain the vascular and anti-ischemic protection role played by PA *before* that an ischemic cerebral accident occurs.

PA is a mild stressor for the organism; causing an increased need for oxygen and energetic substrates, stimulates a series of changes aimed to ensure the amelioration of cerebral vascularization both in quantitative (short-term modification) and qualitative (long-term adaptations) way. The quantitative aspect is what acts in the first play. It is in fact well established that acute exercise increases the cardiac output for adaptation to the increased needs mentioned above, thus leading to a prompt increase in the cerebral vascular flow [123,124,125,126]. Regular PA, on the other hand, can induce increased regional cerebral blood flow by promoting qualitative modifications, angiogenesis and collateral vessels recruitment; the mechanism of recruitment of already existing collateral vessels has a protective effect towards the circle of Willis against high-grade extra or intracranial stenosis or occlusion to reduce risk of watershed infarctions, thus making exercise a modulator factor of cerebral autoregulation [127]. Recently-conducted in vivo animal studies showed how treadmill activity promotes angiogenesis in ischemic stroke animal models [128,129,130]. Pianta et al. [130] found that animals that underwent short durations of exercise (30 m and 60 m exercise groups) prior to stroke induction showed reduced infarct volumes compared to non-exercised stroke animals and histologically, qualitative analysis of surviving cells in the peri-infarct area showed a concomitant increase in the number of live cells as the duration of exercise increased. In this study, the amounts of neurotrophins/angiogenetic factors such as Vascular Endothelial Growth Factor (VEGF), VEGFR-2, and Ang-2 were found to be augmented near the site of the brain injury [130]. It’s supposed that even an early in life PA confers long-lasting protection against cerebral ischemia, as demonstrated by Serra et al. [131], through an epigenetic stable upregulation of the expression of neurotrophic factors which are linked to increased neuronal growth providing the brain with a greater “neuronal reserve”.

Figure 6 summarizes the main anti-ischemic effects of PA.

A neuroradiological study demonstrates that physically active older adults have been found to display a higher number of small cerebral vessels than physically less active older adults [132]. A very recent study conducted in humans has found that a single session of aerobic exercise is adequate to produce transient measurable changes in cerebral perfusion [133] although is unclear if this represents the earliest stages of neural and/or vascular plasticity or a distinct transient mechanism.

Finally, exercise’s anti-ischemic effect may be also linked to the mechanism of preconditioning [134]. Preconditioning is defined as the exposure of a system or an organ to a conditioning stimulus to induce tolerance or resistance to a subsequent injury [135] by triggering cells and organisms to express intrinsic protective factors, thus helping them acquire tolerance and self-defense against later possible damage [136]. Several studies have demonstrated that preconditioning exercise (i.e., prophylactic exercise prior to stroke) provides significant neuroprotection against acute stroke through the promotion of angiogenesis, mediation of inflammatory responses, and inhibition of neuronal apoptosis [134]. In the 2,3,5-triphenyltetrazorlium chloride (TTC) study, the tissue infarction in rats with preconditioning exercise was decreased compared with the brain of rats without preconditioning [134]. Preconditioning with different exercise protocols such as moderate continuous training (MCT) and high-intensity interval training (HIIT) confers similar protection against neurological deficits and tissue injury following an acute stroke [137] although the benefit seems to be directly proportional to the intensity of training [138] and to a longer period of exercising before the ischemic injury [139,140].

The mechanisms through which preconditioning exercise could protect the brain are various. Preconditioning exercise activates astrocytes and improves angiogenesis in the penumbra areas following brain ischemia [141] exerting regulation of multiple factors significant for neurovascular integrity—i.e., endotelin-1, VEGF, IGF-1 and midkine (MK) a heparin-binding growth factor promoting angiogenesis [141,142,143,144,145,146].

Moreover, preconditioning exercise improves different structural and functional components of the blood–brain barrier (BBB) [134] whose dysfunction is one of the key pathobiological mechanisms leading to local and systemic immuno-inflammatory activation and to increase local injury via cerebral edema.

The complex interaction between preconditioning and brain protection towards an ischemic event is shown in Figure 7.

## 6. Is There Any Role for Myokines and Neurotrophins?

A myokine is a peptide or a protein secreted by skeletal muscle cells, either locally (in skeletal muscle interstitium without entering systemic circulation) or in the bloodstream during but also independently of muscle contraction. Since during or after PA various peptides may also be secreted by other organs/tissues, increased circulating levels of a peptide upon muscle activity is not a condition sufficient to establish whether the molecule behaves as a myokine [147]. The skeletal muscle secretome might mediate the effects of exercise on metabolic and cardiovascular health and seems to be involved in the protection against several types of cancer [5]. The overall concept of crosstalk between contracting muscle and distant organs appears to be also likely to be suitable for the brain. The hypothesis is, therefore, that the muscle secretome might be involved in mediating the beneficial effects of exercise on brain health. The proteins with a mainly targeted mechanism of action towards the neural network released in response to exercise are named “neurotrophins”. This is a family of molecules with various neurotrophic activities (stimulation of neuronal survival, growth and/or differentiation) [148], whose levels increase after exercise in several experimental models. In consideration of this general premise, and before addressing in detail the description of what is known today about the role played by muscle myokines and more generically by neurotrophins on the facilitation of neuronal recovery after ischemic damage, we want to establish the three fundamental questions that we will try to answer individually during the discussion of each substance in order to define as clearly as possible the role played by them:(1)If it is true that both in animal models and in human experiments it has been demonstrated that after PA there is an increase in the concentration of different neurotrophins within different structures of the nervous system, is it possible to definitely identify the cells that produce and release them?(2)Are the cells that produce and release neurotrophins located within the nervous system or is there evidence that the muscle itself during contraction or after contraction produces and releases neurotrophins into circulation that would then reach their site of action through the bloodstream?(3)The increased concentration of neurotrophins in the nervous system after exercise can be actually related to their protective role for a good neurological function, thus demonstrating that regular PA through myokines and/or neurotrophins acts in protecting the central nervous system?

Figure 8 shows some of the myokines most closely monitored to date and their potential mechanisms of action in the context of neuroprotection.

### 6.1. Brain-Derived Neurotrophic Factor (BDNF)

BDNF is a neurotrophin involved in regulating the growth and survival of existing neurons and also growth and differentiation of new neurons and synapses, playing an active role in learning and memory; low levels of BDNF have been found in subjects with chronic neurological diseases (e.g., Alzheimer’s disease, depression) and an increase of its plasmatic levels has been demonstrated after acute aerobic exercise in an intensity-dependent manner [7,149,150] and in multiple sclerosis patients as reported by Gold et al. [151] and Castellano et al. [152] (8 weeks of aerobic training at 60% of VO_2_ peak).

Determine with certainty if BDNF is actually released after exercise, which cells produces it and in which entity it is still under investigation today. The main origin of exercise-induced circulating BDNF is likely to be the brain rather than the muscle (specific areas of the human brain such as the hippocampus and cerebral cortex) [150,153,154]. An elegant paper by Rasmussen et al. reveals that the analyses of the arterial-to-internal jugular venous difference in plasma BDNF levels suggest that neurotrophin produced and released from the brain contributes 70–80% of circulating levels following 4 h of aerobic exercise [153]. In this regard, Matthews et al. demonstrate in 2009 that BDNF mRNA and protein expression were increased in human skeletal muscle after exercise, but muscle-derived BDNF appears not to be released into the circulation [155].

Nevertheless, with the uncertainty regarding the main source, exercise-induced BDNF may be reasonably considered as an effective mediator of improved neurological health, especially inducing an exercise-dependent proliferation and growth of hippocampal dentate gyrus cells. The central role of the BDNF is supported by the evidence that blocking its signaling abolished the effects of exercise on synaptic plasticity, learning and memory [78]. Pre-exercised mice seem to reach higher levels of BDNF in ischemic hippocampus than sedentary [156] positively influencing the synaptic plasticity and the dendritic complexity through its receptor Synapsin I, a synaptic mediator for the action of BDNF, increased in proportion to the growth-associated protein (GAP-43) and its signal transduction receptor (trkB) [157]. Interestingly exercise intensity does not influence the extent of the PA-induced neuroplasticity [158].

Winter et al. [77] observed increased BDNF level in humans after a single bout of running at a high intensity. The magnitude of BDNF increase after an acute bout seems to be exercise-intensity dependent [159,160]; BDNF levels decrease to baseline levels within minutes to several hours [151,161]. Schmidt-Kassow et al. [162] found that BDNF reaches its maximum serum concentration after 20 min of exercise (mainly high-intensity aerobic training) and returns to baseline after approximately 10 min of recovery. No sex differences during baseline or recovery are reported by these authors although the increase in the BDNF concentration during the exercise phase appears to be more pronounced in men than in women. The rapidity with which the plasma concentration of BDNF decreases after the cessation of exercise has been suggested by some authors as a rapid plasma clearance of the neurotrophin by the target tissues [163]. Similar results are reported by Yarrow et al. in healthy untrained college-aged males who underwent a five-week traditional or eccentric-enhanced progressive resistance training intervention [164].

Changes in BDNF levels after regular aerobic training are also reported by Seifert et al. [150] (12 healthy sedentary males, thhree months of endurance training) and Zoladz et al. [165] (13 young, healthy physically active men, five weeks of moderate-intensity endurance cycling training program), but not by Ruscheweyh et al. [166] (62 healthy elderly individuals, six months medium or low-intensity PA) and Shulz et al. [167] (28 patients with multiple sclerosis, eight-week aerobic bicycle training at 60% VO(2)max).

Almost all studies that have tried to demonstrate an increase in BDNF plasma levels after resistance training have failed to provide convincing results. Correia et al. [168] (acute resistance training in healthy subjects); Goekint et al. [169] (fifteen untrained subjects-strength training program for 10 weeks); Schiffer et al. [170] (27 healthy students, 12-week training intervention, moderate endurance training and strength training with high loads); Levinger et al. [171] (49 subjects with various degree of metabolic risk factors, ten weeks of resistance training).

As a result of these conflicting results, given also the uncertain crossing of the BDNF through the BBB [172] some authors hypothesize that PA may indirectly stimulate the release of hippocampal BDNF. The ketone β-hydroxybutyrate (DBHB), produced in the liver and permeable to the BBB, has been shown to accumulate and stimulate the production of BDNF in the hippocampus following exercise [173]. This finding, replicated by direct ventricular infusion of DBHB, would seem to associate the central production of BDNF with metabolic-related factors. Moon et al. [174] assumes that PA, through the intervention of the myokine cathepsin B, may promote hippocampal expression of BDNF. They found, in mice, after running, an increase in expression of Ctsb gene (whose transcription leads to the synthesis of cathepsin B) and a subsequent increase in plasmatic levels of this myokine that is also able to cross the blood-brain barrier. Other reports [175] seem to associate the hippocampal activation of the BDNF with another myokine, Irisin, whose release into circulation after exercise, however, is currently subject to conflicting reports [53]. A meta-analysis of 29 studies (N = 1111 participants) examining the effect of exercise on BDNF levels after a single session of exercise, after a session of exercise following a program of regular PA and evaluating resting BDNF levels following a program of regular PA demonstrated a moderate effect size for increases in BDNF following a single session of exercise; the effect is intensified after regular PA but the overall increase in plasmatic levels of BDNF after regular PA is small. The magnitude of these effects seems to be lower in females than in males [176]. Another more recent meta-analysis involving 29 studies found similar results confirming that aerobic but not resistance training interventions increased resting BDNF concentrations in peripheral blood without significant difference between males and females [177].

In conclusion, there is little evidence that muscle-derived BDNF is released into the bloodstream from muscle during or after PA whereas exercise-induced BDNF comes from different tissues, first of all, the hippocampus; there is no evidence that BDNF may mediate the muscle-brain cross-talk; there is good evidence that acute and chronic aerobic exercise induces a local (mainly hippocampal) and systemic release of BDNF with effects mainly on cognitive functions and in neuroplasticity; no data is available to date regarding a possible increased post-ischemia recovery capacity of the neuronal structures after regular PA directly involved in the increased (local or systemic) availability of BDNF.

### 6.2. Insulin Growth Factor (IGF)-1

Another recently discovered neurotrophin is the Insulin Growth Factor (IGF)-1. Numerous studies support a critical role for IGF-1 in normal brain development and function [178]. Induced by circulating growth hormone (GH), IGF-1 is primarily produced by the liver before being released into the circulatory system. Various reports indicate that IGF-1 is released by other tissues including the skeletal muscle [93,179,180,181] playing an active role in mediating the recovery of neuronal function through the PA together with BDNF, despite an excessive release of IGF-1 is reported to have opposite effects, inducing an increased level of neuroinflammation and the worst outcome [182]. The attempt to assess if acute exercise effects peripheral IGF-1 levels in humans lead to conflicting results [183,184], although Rojas Vega et al. reported a plasmatic increase of IGF-1 (but not of BDNF) levels after a single bout of resistance training [185], Sillanpaa et al. [186] report an 8% (p = 0.097) increase after 21 weeks of endurance training in healthy old women and Cappon et al. [187] showed that 10-min above-lactate threshold cycle ergometer exercise led to a brief and small increases in circulating IGF-I that were independent of circulating GH, other authors failed to obtain similar results [152,169,170,188].

The effect on brain neurogenesis is altogether achieved by indirect actions of IGF-1 on other neurotrophins, such as BDNF, VEGF, or Growth Hormone (GH) [189], also due to the fact that circulating IGF-1, unlike other neurotrophins, crosses the BBB. In this regard, IGF-1 is widely expressed centrally, with special reference to the dentate gyrus, the only hippocampal subregion that supports neurogenesis in the adult brain. [190,191,192]. The close interaction between different neurotrophic factors seems to be a typical feature of increased neuroplasticity due to exercise, as for example demonstrated by exercise-induced paralysis recovery of an experimental model of brain ischemic rats, in which the interaction of GAP-43 phosphorylated at serine 41 (pSer41-GAP-43), calmodulin, PK-C and nerve growth factor (NGF) is able to grant neurite formation, synaptic connections, and neuronal remodeling [193]. The close functional interconnection between some (but not all) neurotrophic factors is demonstrated by the experimental finding that Blocking hippocampal uptake of circulating IGF-1 or the IGF-1 receptor significantly blunts the exercise-induced increase in BDNF mRNA and protein, as well as that of it precursor protein [80,194].

Very interesting are the results of some studies that have evaluated the role of IGF-1 in experimental models of hypoxia. IGF-1 is strongly expressed in damaged brain tissue after a hypoxic-ischemic injury and exogenous administration of IGF-1 shortly (hours) after the acute injury has been shown to be neuroprotective [195]. A single dose of IGF-1 (50 microg) administered intracerebroventricularly 2 h after the hypoxia injury in rats demonstrated to reduce the extent of cortical infarction after 5 days and 20 days, significantly reducing the percentage of selective neuronal loss (P = 0.027), positively influencing both the extent of cortical infarction and ongoing progressive neuronal death during brain recovery from hypoxic-ischemic injury [196]. Some of IGF-1 central actions could be mediated through the N-terminal tripeptide fragment of IGF-1: Gly-Pro-Glu (GPE) in rats [197]. In animal models, IGF-1 treatment increases oligodendrocyte due to suppression of apoptosis and partly to increased proliferation and increases reactive glia only improving proliferation. [198] and ameliorates age-related declines in the performance of tasks involving working memory [199]. Speculatively, taken together, these effects may allow us to hypothesize a multi-level IGF-1 mediated neuroprotection.

### 6.3. Irisin

Irisin, first reported in 2012 [200], is a myokine derived by the cleavage of the membrane protein fibronectin type III domain-containing protein 5 (FNDC5), whose expression and transcription is under the close control of the metabolic mediator peroxisome proliferator-activated receptor gamma coactivator 1 alpha (PGC-1α). Irisin is involved in thermogenesis, lipid metabolism and obesity reports [53], and exercise, trough irisin release, increases hippocampal expression of BDNF [201]. Li et al. [202], furthermore suggest that irisin decreases ischemia-induced neuronal injury by activating Akt and ERK1/2 signaling pathways and thus contributes toward neuroprotective effects of exercise against cerebral ischemia. Similar results are achieved by Peng et al. [203] demonstrating that irisin mitigates oxygen-glucose deprivation-induced neuronal injury in part by inhibiting the ROS-NLRP-3 (reactive oxygen species-Nod like receptor pyrin-3) inflammatory signaling pathway. Finally, Irisin could contribute to the anti-inflammatory effects of exercise: blunts the release of IL-6 and IL-1β from cultured astrocytes and reduced expression of COX-2 (cyclooxygenase-2) and phosphorylation of the PI3K (phosphatidylinositol 3-kinase) and Akt (protein kinase B) signaling pathway [204].

### 6.4. Other Neurotrophic Factors

In addition to BDNF and IGF-1, exercise also regulates the expression of VEGF responsible for endothelial cell proliferation and angiogenesis and for securing neurotrophic, neuroprotective, and neurogenic effects [205]. Exercise-induced neurogenesis and brain micro-angiogenesis is associated with an increase in VEGF [145,206], the neurotrophin ciliary neurotrophic factor (CNTF) [207], and the Fibroblast Growth Factor 21 (FGF21) [208].

Release of Neural growth factor (NGF) is also described but appears to be less associated with both acute and regular PA. [209]. Local and temporal differences in NGF versus BDNF upregulation in response to exercise have been observed [210]. Neurotrophin 4 is another muscle-derived factor acting as a neurotrophin able to induce growth and remodeling of adult motor neuron innervation [211].

## 7. Conclusions

Several recent meta-analyses show robust epidemiological evidence indicating that regular PA is associated with a reduced incidence of the main chronic pathologic conditions and of cardiovascular and cerebrovascular events. In this review, we tried to highlight the many contributions of acute bouts of exercise and regular PA in neuroprotection with a focus on cerebral acute ischemia. Our review has some limitations: its approach as a narrative review has implicit the awareness in those who write it and in those who read it that not all the existing knowledge on the subject has been reported, but rather the authors have tried to “ narrate”, precisely, to the best of their ability what is known today on the subject. It should be emphasized that the information about myokines and neurotrophins are subject to constant updates that also lead to review critically and deny concepts that so far have been given as certain or at least likely.

Effects of exercise on neuroplasticity and cognition have been shown across the lifespan from childhood to old age. The results of available data suggest that the functions undergoing developmental changes (e.g., executive functions in childhood and old age) and functions which show a high intraindividual variability across the lifespan (e.g., memory) might benefit most from changes in the central nervous system induced by exercising. The mechanisms underlying this phenomenon are polymorphous, complex and only partially known. Regular PA exerts anti-ischemic effects: neo-angiogenesis and collateral vessel recruitment are the main features of a whole comprehensive mechanism of cerebral preconditioning. Several pieces of experimental evidence confirms that an ischemic cerebral event occurring in the brain of a trained subject can be better managed by cerebral vascular structures and neurons if it occurs in untrained. Furthermore, after brain injury following ischemic stroke, exercise is considered an effective and feasible rehabilitation strategy for improving cognitive and motor functional recovery through the facilitation of neuroplasticity such as through increases in neuronal activity and the potentiation of postsynaptic excitation, as well as enhancements in axonal myelination following ischemic stroke.

Ischemic stroke has as crucial elements thrombosis and inflammation; these two strictly connected processes interact at various points contributing to brain injury and stroke progression. The systemic anti-inflammatory effect of exercise is confirmed by several lines of evidence and sustained by different mechanisms many of which are associated with the reduction of the quantity and the qualitative variation of the fat mass. An intense debate has driven the discussion about the possibility that the anti-inflammatory effect of exercise can also be mediated (to some extent) by myokines or other molecules produced by the muscle during prolonged contraction and released into the bloodstream. The issue” inflammation and exercise” is one of the most interesting and controversial. In contrast to numerous experimental evidence that has confirmed that regular exercise exerts a demonstrable anti-inflammatory effect, much less certain are the mechanisms through which this is achieved. The role of myokines, and therefore the possibility that the muscle performs a complex and intricate endocrine action capable of influencing the structure and function of distant organs and tissues has been proposed in the last decade as a result of the growing information collected in this direction. We have extensively discussed the pros and contras of this fascinating hypothesis elsewhere [5,7,53,64]. Certainly, the lack to date of experimental data that has evaluated the anti-inflammatory effect of exercise in randomized controlled trials that have enrolled patients with one or more chronic diseases limits our knowledge about the issue of inflammation and exercise. How important is the reduction of systemic inflammation level in mediating the beneficial effects of regular exercise in patients with multimorbidity and through which mechanisms regular PA is able to determine an anti-inflammatory effect we do not yet know with certainty.

The neuroprotective role of muscular myokines and more broadly of the neurotrophins is under close investigation: several authors have released good experimental evidence in mice in which relatively low levels of exercise are able to ensure the upregulation of the expression in various brain regions of a range of neurotrophins associated with synaptic plasticity: IGF-1, synapsin-I and BDNF [93,212]. Data obtained in humans suggests that many of the protective benefits of regular PA are obtained through elevations in central and peripheral neurotrophic factors. Increases in plasma levels of neurotrophins (mainly BDNF) have been reported to be associated with both acute sessions of high-intensity exercise and regular aerobic (but not resistance) PA [177]. The number of muscle-derived factors with hypothetical neurotrophic action is always growing together with the hypothesis about their complex interactions and cross-talk between muscle and brain. In conclusion, the multiple neuroprotective effects of regular PA that we have described in our review, with particular reference to neuroprotection against ischemic damage, are mediated by a complex network of mechanisms triggered by training. As in other areas of research, the translation to humans of experimental evidence obtained with animals is not always immediate, leaving some questions open. The therapeutic potential related to this area of research is considerable: identifying systemic mediators induced by exercise that can protect neurons from ischemia by raising the threshold of ischemic suffering and/or accelerating post-ischemic recovery would allow the clinician to acquire therapeutic weapons in an area (acute cerebral ischemia) in which there are currently no drugs with neuroprotective action. We expect further studies in the near future, both in the field of pathophysiology of cerebral ischemia and in the area of systemic effects of the exercise from which we expect progress potentially able to open up therapeutic intervention scenarios unimaginable to date.

## Figures and Tables

**Figure 1 ijms-21-09086-f001:**
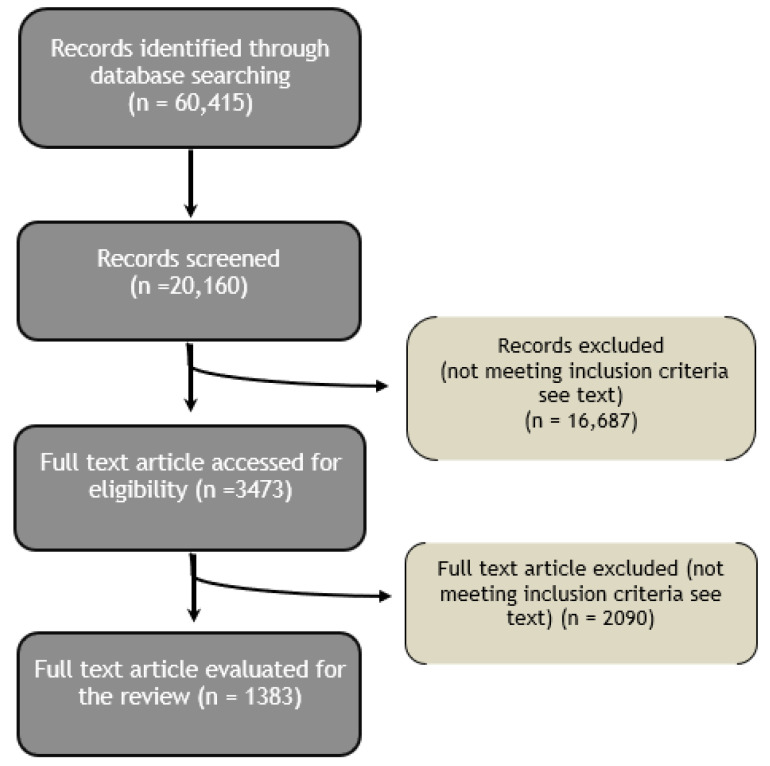
Flow chart of the literature search and selection.

**Figure 2 ijms-21-09086-f002:**
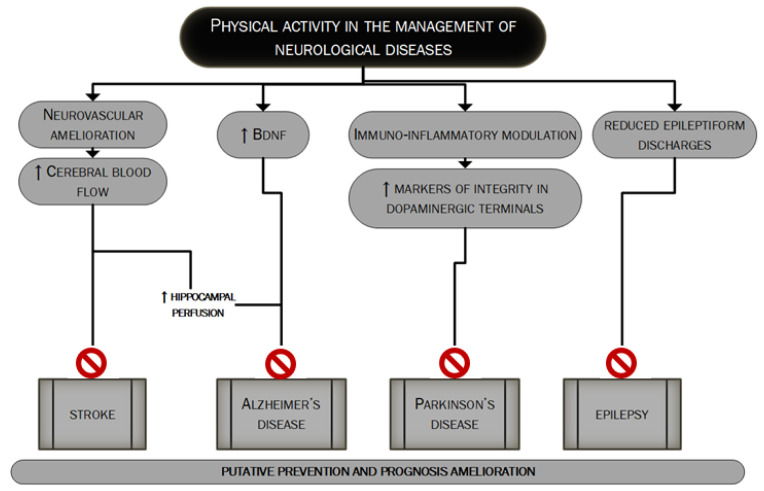
The putative role of PA in the management of major neurological diseases. In this figure, we summarize the main mechanisms through which regular PA could lead to better management of the four major neurological diseases: stroke [127–131], Alzheimer’s disease, Parkinson’s disease and epilepsy [5,21–23]. PA’s role could be pivotal in the prevention as well as in the prognosis’ improvement.

**Figure 3 ijms-21-09086-f003:**
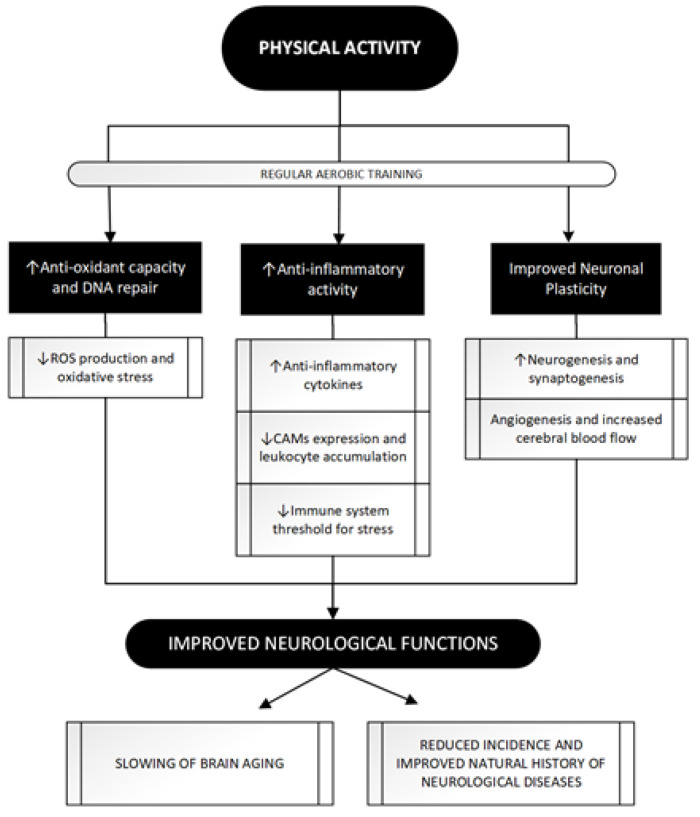
The attempt to improve neurological functions through regular PA: a three paths journey. PA achieves beneficial effects on brain function through its impact in these three main pathways whose modulation eventually leads to a neuroprotective effect. Regular aerobic training as opposed to acute bouts of exercise, seems to be more related to PA positive impact leading to antioxidant [47,48] and anti-inflammatory [51–53,56] effects and, through molecular and vascular changes, shapes neuronal and vascular plasticity [70,75,78,79]. See text for further information. ROS, Reactive Oxygen Species; CAMs, Cell Adhesion Molecules; DNA, DeoxyriboNucleic Acid.

**Figure 4 ijms-21-09086-f004:**
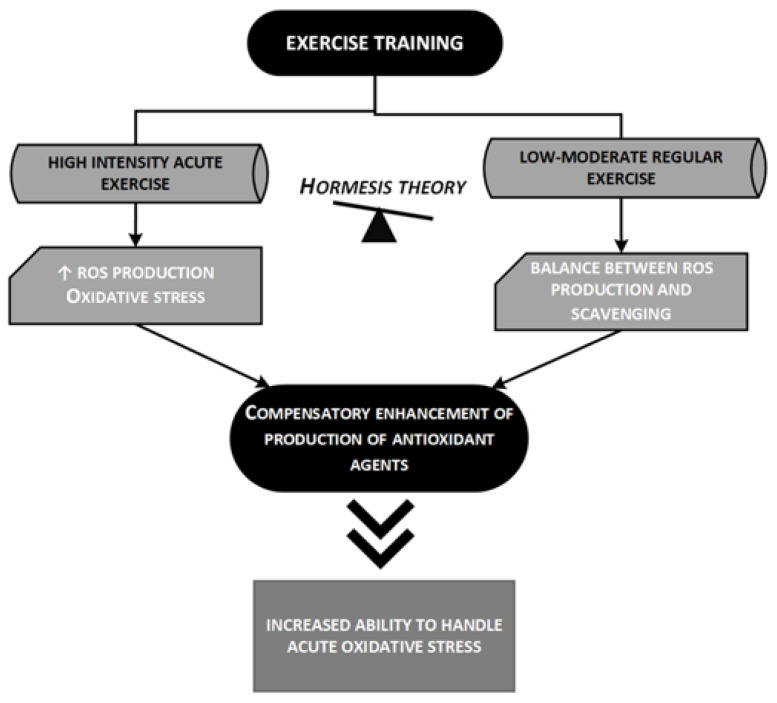
The hormesis theory and the improved antioxidant activity of trained subjects [49,50]. The hormesis theory tries to explain the apparent paradox related to the relationship between oxidative stress and exercise (see text for more details); low-moderate regular exercise enhances endogenous antioxidant capacity resulting in an increased ability to handle acute oxidative stresses. ROS, reactive oxygen species.

**Figure 5 ijms-21-09086-f005:**
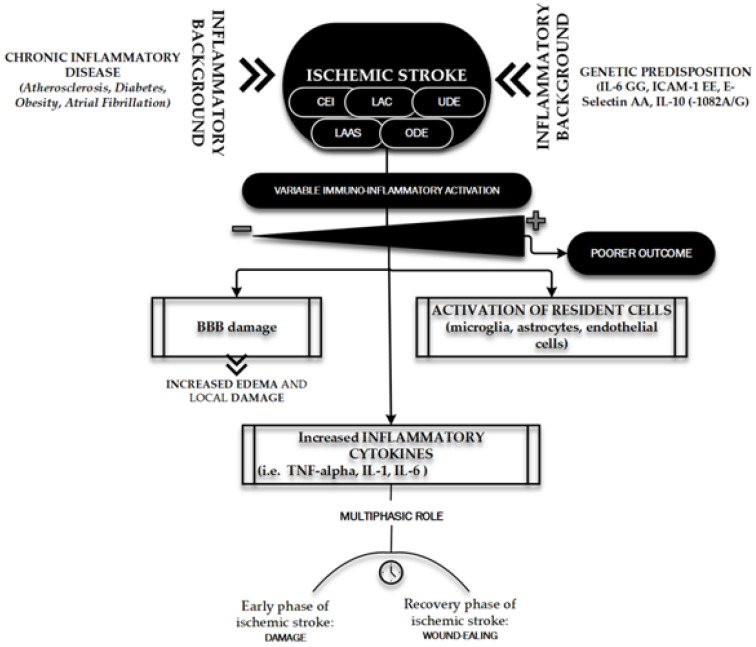
Immuno-inflammatory activation during ischemic stroke. The figure briefly outlines the main determinants and characters involved in the “inflammatory face” of ischemic pathobiology. Chronic inflammatory diseases [98] and genetic predisposition [98,118] represent the key elements playing a leading role in determining the variability of immune-inflammatory activation [98,101,104,107–113] on which depends the patient’s outcome: the inflammatory background elements and the ischemic stroke subtypes (TOAST classification) [121]. CEI, CardioEmbolic Infarct; LAC, LACunar infarct; UDE, a stroke of UnDetermined Etiology; ODE, a stroke of Other Determined Etiology; LAAS, Large Artery AtheroSclerosis; IL, Interleukin; ICAM, Intercellular Cell Adhesion Molecule; TNF, Tumor Necrosis Factor.

**Figure 6 ijms-21-09086-f006:**
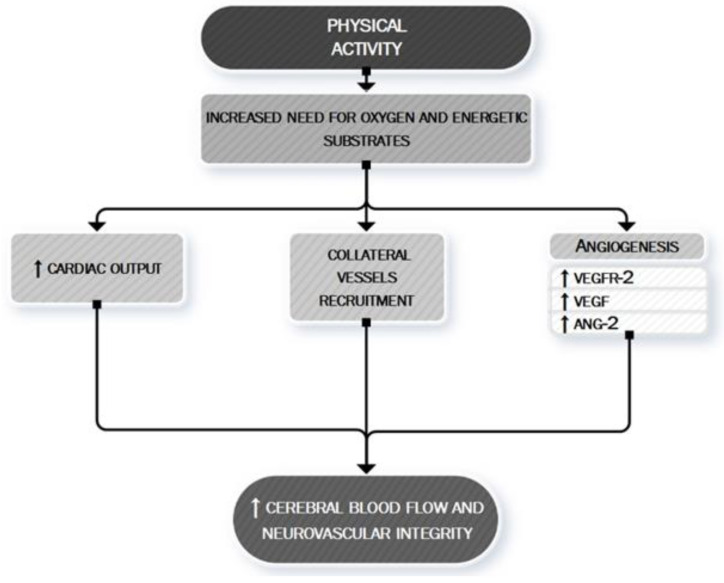
PA: a neurovascular booster. During PA, the increased need for oxygen and energetic substrates leads to the activation of many mechanisms whose ultimate effect is to determine an increased cerebral blood flow and to maintain neurovascular integrity. This aim is exerted both by asserting a boosting of cardiac contractility (quantitative effect) [123–126] and by a local cerebral modulation of neurovascular dynamic (qualitative effect) through the recruitment of collateral vessels [127] and the stimulation of angiogenic processes [130]. VEGF, vascular endothelial growth factor; VEGFR-2, vascular endothelial growth factor receptor 2; ANG-2, Angiopoietin-2.

**Figure 7 ijms-21-09086-f007:**
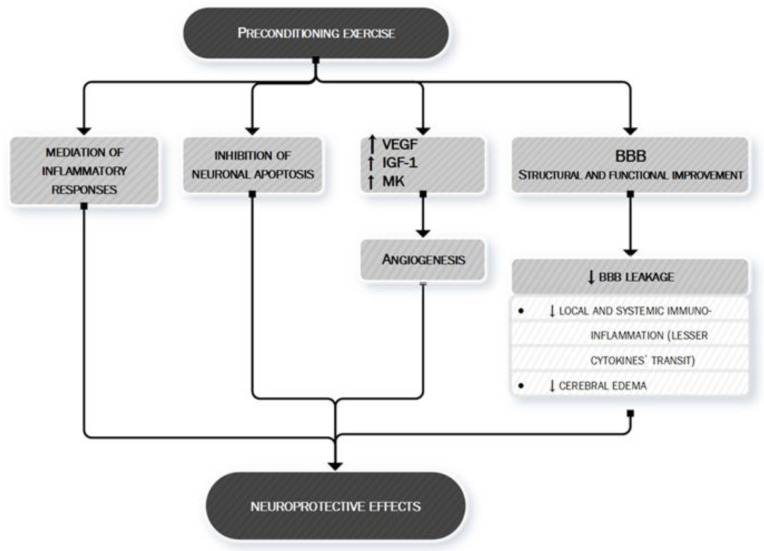
Preconditioning exercise: brain’s eustress. PA is a mild stressor for the organism. In particular, preconditioning exercise (see text for more information) can be defined as an “eustress” because it triggers cells to express multiple factors, thus helping the organism to acquire tolerance and self-defense against later possible damage, such as ischemic stroke. Preconditioning exercise triggers the brain to activate vascular and inflammatory mechanisms (seen in the figure) [134,141–146] whose production remodulates cerebral neurovascular structure so that later pathologic stimuli are less likely to act or less likely to exert important damage. Preconditioning exercise improves the function and structure of blood–brain barrier (BBB) [134] thus limiting the leakage of both fluids (edema) and proteins (cytokines) who play an important role in ischemic stroke pathobiology. VEGF, vascular endothelial growth factor; IGF-1, Insulin-like Growth factor 1; MK, midkine; BBB, blood–brain barrier.

**Figure 8 ijms-21-09086-f008:**
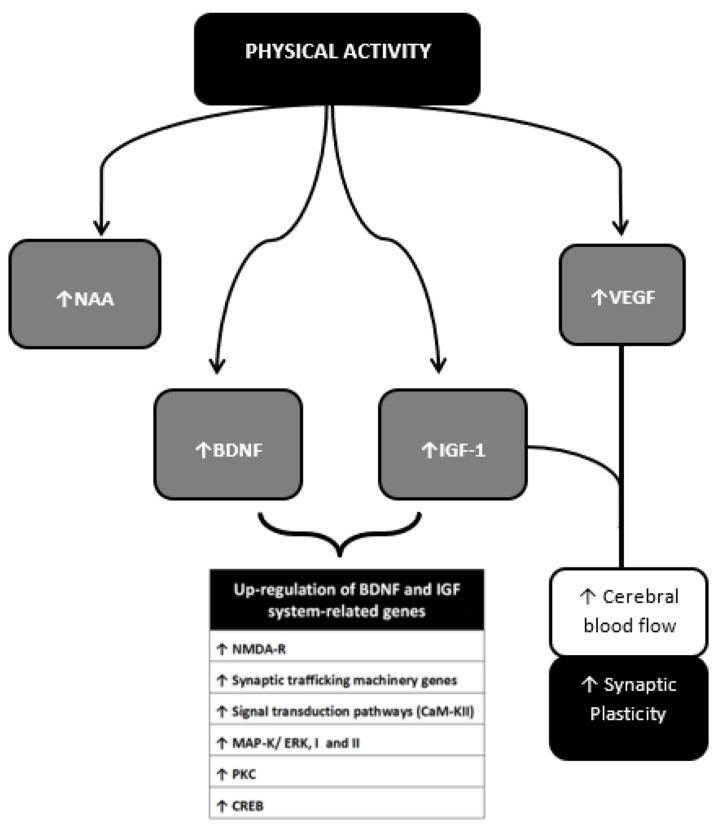
Neurotrophins, Myokines and other muscle-derived factors involved in neuroprotection. Neuroprotective effects of PA result from multiple molecular changes operating directly in the cerebral tissue and indirectly via vascular modulation. These molecular targets interplay with each other achieving different outcomes. The BDNF-IGF1 network [89–194] and the VEGF-IGF1 network represent two main examples. BDNF-IGF1 network mediates the upregulation of genes mainly related to neuronal viability [78,79]. See text for further information. NAA, N-acetylaspartate; BDNF brain-Derived Neurotrophic Factor; IGF-1, Insulin-Like Growth Factor 1; VEGF, Vascular Endothelial Growth Factor; GABA, Gamma-AminoButyric Acid; NO, Nitric Oxyde; NMDA-R, *N*-methyl-d-aspartate receptor; CaM-KII, calcium/calmodulin protein kinase II; MAP-K/ERK I and II, mitogen-activated protein kinase; PK-C, protein kinase C; CREB, cAMP response element-binding protein.

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
