# Peer review of "Role of Regular Physical Activity in Neuroprotection against Acute Ischemia"

_ijms, 2020, doi:10.3390/ijms21239086_

Round 1
Reviewer 1 Report
This paper provides a concise and thorough summary of the current literature of neurotrophins and their relationship with PA.
The figures are easy to follow, and the manuscript is well written.
The following are some issues that I found of concern.
Line 30-32 state that this is a review paper. It would be wise to define whether this was (presumably) a narrative review paper or a systemic review paper.
Following the fact that the authors intended this to a be a review paper, there some parts that should be added in the methodology section.
Data sources and research
Line 38-39: From what year was the MEDLINE search performed
Why was only MEDLINE used for review (the reason for excluding other well - known databases ?)
Would the authors be content to retrieve references from just one single database?
The reasons justifying this decision should be stated.
Line 47: The only exclusion criteria stated by the authors were those limited to the English literature. Can the authors be more specific on the criterion to exclude those unsuitable for their review?
Also, as in most review papers, a flow diagram on the process of the literature selection would be helpful to the readers.
Line 49-50 Each single participant-> so how many in total were allocated to the literature search?
Following the introduction, the authors described their results into the following sections supported by
figures.
- Neuroprotective effects of regular PA
- Immunoinflammatory activation during and after the acute phase of ischemic stroke
- Anti-ischemic effects of PA
- Any role of myokines or neutrophins?
Figures
I found the figures concise but difficult to follow
Figure 1 , Figure 4 -> they should be cited in the manuscript
Figure 5-> there are 2 figures for Figure 5, and they are not cited in the manuscript
Figure 6-> not cited in the manuscript (and should be figure 7)
As such, they should be cited and to match with the content of the manuscript, I would suggest either of the following
- To cite the references within the figures
- Or to produce some tables that could summarize the results of the studies. Based on the formats the authors have presented,it would be helpful if the authors present some tables from the 4 sections above and provide some summary of the findings, as provided in other narrative review papers.
Line 683 Conclusions
Some limitations on the review process should be stated in conclusion.(one of them being of course, that this is not a quantitive systemic review)
Line 706-709 : Although the authors present some present debate, “about the anti-inflammatory effect of exercise mediated to some extent by myokines”, some further discussion on what points are agreed or disagreed based on their review of the literature should be stated in more detail, based on their results from the reviews.
Line 719-720 The authors conclude the paper by stating that “data in humans are scares and it is still too early to indicate their real clinical evidence ". Still, after the review, it would be advisable to conclude the paper in a more optimistic tone by presenting the authors’ thoughts on what steps should be taken in order to improve the future potential or clinical benefits that are challenging at this present moment.
After a thorough review and suddenly concluding in a negative tone, does little to convince the readers. I would advice to reword the last paragraph.
Author Response
REVIEWER #1
We would like to thank you for the detailed review of our manuscript. We greatly appreciate the effort you made concerning your critique for the review of our study. We have accepted all your suggestions and revised the article according to them.
Line 30-32 state that this is a review paper. It would be wise to define whether this was (presumably) a narrative review paper or a systemic review paper.
Following the fact that the authors intended this to a be a review paper, there some parts that should be added in the methodology section.
Thank you for your kind suggestion, the term “narrative review” is now replaced in the abstract. We thank the reviewer for allowing us to clarify in more detail how the literature search was conducted in our review.
Data sources and research
Line 38-39: From what year was the MEDLINE search performed
The time interval for which the search was performed (between 2000 and 2020) has been added to the text
Why was only MEDLINE used for review (the reason for excluding other well - known databases ?). Would the authors be content to retrieve references from just one single database?
The reasons justifying this decision should be stated.
Thank you. This is an interesting point. Medline is certainly one of the most important databases in the world that many researchers rely on as the first (and often the only) source of literature research. The search and comparison in multiple databases would have involved first of all a lengthening of the time of research and data analysis for the avoidance of duplicates (which would have been the vast majority). In our experience, and this is our opinion, the research carried out on a single database of great quality and international recognition is adequate to ensure an appropriate level of accuracy regarding the number and quality of the articles evaluated. As suggested by the reviewer the reasons of our choice are now briefly reported in the methodology section.
Line 47: The only exclusion criteria stated by the authors were those limited to the English literature. Can the authors be more specific on the criterion to exclude those unsuitable for their review?
Thank you for the suggestion. More detailed exclusion criteria have been added to the text.
Also, as in most review papers, a flow diagram on the process of the literature selection would be helpful to the readers.
Thank you for your kind suggestion, we added in the revised version a flow diagram explaining the process of the literature search and selection.
Line 49-50 Each single participant-> so how many in total were allocated to the literature search?
The literature consultation and the selection of the articles was carried out by all the authors. This statement has been modified in the revised version.
Following the introduction, the authors described their results into the following sections supported by figures.
- Neuroprotective effects of regular PA
- Immunoinflammatory activation during and after the acute phase of ischemic stroke
- Anti-ischemic effects of PA
- Any role of myokines or neutrophins?
Figures
I found the figures concise but difficult to follow
Figure 1 , Figure 4 -> they should be cited in the manuscript
Figure 5-> there are 2 figures for Figure 5, and they are not cited in the manuscript
Figure 6-> not cited in the manuscript (and should be figure 7)
Following your kind suggestion in the revised version the figures (and their citation in the text) have been modified following the proper indications of the reviewer
As such, they should be cited and to match with the content of the manuscript, I would suggest either of the following
To cite the references within the figures
Or to produce some tables that could summarize the results of the studies. Based on the formats the authors have presented, it would be helpful if the authors present some tables from the 4 sections above and provide some summary of the findings, as provided in other narrative review papers.
As suggested by the reviewer, we have implemented the tables adding references, when appropriate.
Line 683 Conclusions
Some limitations on the review process should be stated in conclusion.(one of them being of course, that this is not a quantitive systemic review)
Thank you for the suggestion. We are fully aware of the limitations that are involved in the choice to write a "narrative review" and we have reported this in our revised conclusions.
Line 706-709 : Although the authors present some present debate, “about the anti-inflammatory effect of exercise mediated to some extent by myokines”, some further discussion on what points are agreed or disagreed based on their review of the literature should be stated in more detail, based on their results from the reviews.
The issue” inflammation and exercise” is one of the most interesting and controversial. In contrast to numerous experimental evidence that has confirmed that regular exercise exerts a demonstrable anti-inflammatory effect, much less certain are the mechanisms through which this is achieved. In the revised version we try to provide some conclusive messages on the subject, indicating that there is still a substantial lack of experimental evidence on this topic, especially in patients with multimorbidity in which it is more difficult to perform exercise at high intensity and for prolonged lengths of time and who often have a condition of sarcopenia that limits the capacity and function of the muscle.
Line 719-720 The authors conclude the paper by stating that “data in humans are scares and it is still too early to indicate their real clinical evidence ". Still, after the review, it would be advisable to conclude the paper in a more optimistic tone by presenting the authors’ thoughts on what steps should be taken in order to improve the future potential or clinical benefits that are challenging at this present moment.
After a thorough review and suddenly concluding in a negative tone, does little to convince the readers. I would advice to reword the last paragraph.
We agree with the reviewer that maybe the last paragraph of the conclusions does not adequately reflect the complexity of the topic we have dealt with and the considerable potential for both scientific and therapeutic research related to it. The last paragraph has been totally rewritten in the revised version of the review.
We hope that we have successfully changed our manuscript according to your suggestions and that we have provided all the necessary explanations. We also hope that the manuscript now fulfills your criteria, and the Journal criteria for publication.
Reviewer 2 Report
This review summarized about the neuroprotective and anti-ischemic role of regular exercise with the possible role of neurotrophins and myokines. The content of this manuscript is considered to be a very important part in understanding the mechanisms for functional improvement of regular exercise, and it will be able to be cited in many future studies.
It is necessary to correct the part that repeats the full term summarized in the abbreviation described in the part such as “Physical Activity (PA).” In addition, it is necessary to correct the use of capital letters in the paragraph title such as 3. Neuroprotective effects of Regular Physical Activity,” and the use of capital letters in the figure needs to be corrected. This part needs to be revised in accordance with the Introduction of authors of IJMS.
Author Response
We would like to thank you for your expert and meticulous review of our manuscript. Thank you very much for your positive opinion regarding our manuscript. We put a lot effort in this study and we appreciate your opinion very much.
1.This review summarized about the neuroprotective and anti-ischemic role of regular exercise with the possible role of neurotrophins and myokines. The content of this manuscript is considered to be a very important part in understanding the mechanisms for functional improvement of regular exercise, and it will be able to be cited in many future studies.
It is necessary to correct the part that repeats the full term summarized in the abbreviation described in the part such as “Physical Activity (PA).” In addition, it is necessary to correct the use of capital letters in the paragraph title such as 3. Neuroprotective effects of Regular Physical Activity,” and the use of capital letters in the figure needs to be corrected. This part needs to be revised in accordance with the Introduction of authors of IJMS.
We thank the reviewer for the suggestions regarding the need to increase attention to the use of abbreviations and the use of capital letters according to the instruction for authors. We have modified the manuscript accordingly.
We hope that we have successfully changed our manuscript according to your suggestions and that we have provided all the necessary explanations. We also hope that the manuscript now fulfills your criteria, and the Journal criteria for publication.
Round 2
Reviewer 1 Report
The authors delivered a thorough revision.
All potential issues raised were answered and reflected appropriately.